# Molecular Breeding and Drought Tolerance in Chickpea

**DOI:** 10.3390/life12111846

**Published:** 2022-11-11

**Authors:** Ruchi Asati, Manoj Kumar Tripathi, Sushma Tiwari, Rakesh Kumar Yadav, Niraj Tripathi

**Affiliations:** 1Department of Genetics & Plant Breeding, College of Agriculture, Rajmata Vijayaraje Scindia Krishi Vishwa Vidyalaya, Gwalior 474002, India; 2Department of Plant Molecular Biology & Biotechnology, College of Agriculture, Rajmata Vijayaraje Scindia Krishi Vishwa Vidyalaya, Gwalior 474002, India; 3Directorate of Research Services, Jawaharlal Nehru Agricultural University, Jabalpur 482004, India

**Keywords:** abiotic stress, candidate genes, drought tolerance, crop improvement, climate change

## Abstract

*Cicer arietinum* L. is the third greatest widely planted imperative pulse crop worldwide, and it belongs to the Leguminosae family. Drought is the utmost common abiotic factor on plants, distressing their water status and limiting their growth and development. Chickpea genotypes have the natural ability to fight drought stress using certain strategies viz., escape, avoidance and tolerance. Assorted breeding methods, including hybridization, mutation, and marker-aided breeding, genome sequencing along with omics approaches, could be used to improve the chickpea germplasm lines(s) against drought stress. Root features, for instance depth and root biomass, have been recognized as the greatest beneficial morphological factors for managing terminal drought tolerance in the chickpea. Marker-aided selection, for example, is a genomics-assisted breeding (GAB) strategy that can considerably increase crop breeding accuracy and competence. These breeding technologies, notably marker-assisted breeding, omics, and plant physiology knowledge, underlined the importance of chickpea breeding and can be used in future crop improvement programmes to generate drought-tolerant cultivars(s).

## 1. Introduction

Chickpea is a diploid annual crop that is extremely self-pollinated [1]. After the faba bean and field pea, it is the world’s third most significant pules crop [2]. It is a popular cool-season legume crop with a 738-megabyte genome size [3]. With an annual production of 10.13 million tonnes from a land area of 9.44 million hectares and a productivity of 1073 kg ha^−1^, India is the greatest producer of chickpeas in the world [4]. Chickpeas are grown in 52 countries, together with Africa, Asia, Australia, and South Europe [5]. Mexico, Turkey, Canada, Iran, Australia, Tanzania, Ethiopia, Spain, and Burma are also notable producers of chickpea. Its seeds come in two varieties. The ‘desi’ chickpea is hardy in character, while the Kabuli chickpea has a delicate seed coat and appears to have evolved from the desi varieties [6,7]. In semi-arid zones, chickpea is cultivated in the form of a dry weather crop [8]; however, in cold climatic zones, it is grown as a rainfed crop [9,10]. In actuality, about 90% of the chickpea crop is cultivated in a rainfed environment [11,12,13]. Without irrigation, the crop is affected [14] at vegetative as well as reproductive phases. After illnesses, drought is the second most significant constraint to the yield of chickpea crop [15]. Drought has been reported as a factor of 40–50 percent yield reduction in chickpea [11,12,16,17,18].

The chickpea is also termed as the “poor man’s meat” [19], since it is important for supplying protein sources [20]. Nutritionists have also highlighted its importance due to high nutritional contents in it [21]. Chickpea is high in lysine and arginine [22], but low in methionine and cystine [23]. In general, the Kabuli type contains more protein than the desi kinds. It contains more calcium and phosphorus than most other pulse crops [24,25]. Chickpea seeds comprise 23% protein, 64% total carbohydrates (47% starch, 6% soluble sugar), 5% fat, 6% crude fibre, and 2% ash on average, as well as micronutrients, for example phosphorus, calcium, magnesium, iron, and zinc [26]. Recently, Singh et al. [21] also reported chickpea as good source of Fe and Zn. Consequently, Samineni et al. [27] examined the effects of drought stress on nutritional parameters of chickpea and observed significant differences in the nutritional contents due to stress.

Chickpea is mostly cultivated in the post-rainy season [28], using soil moisture that has been retained from the previous rainy season [29]. As a result, the crop is frequently subjected to severe heat and drought pressure [12,13,22]. Drought, among other abiotic factors, has a significant impact on chickpea output [30]. Drought and heat stress have been reported to have reduced chickpea yields by about 50% due to the damaging effects of the membrane and reduced photosynthesis [31].

The four climatic elements that are changing will have an impact on how much water plants consume [32]. These elements include rising CO_2_ concentrations and temperatures, more erratic precipitation, and changes in humidity. Due to the increased variability in precipitation during the growing season and more so in soils with low water holding capacity, these climate changes may result in an increase in the atmospheric water demand by crops and an increase in the potential for limitations in the availability of water in the soil. In the long run, breeding cultivars with high water use efficiency (WUE) is a more realistic and cost-effective strategy for raising yields in drought-prone locations. WUE promotes modest water absorption while maintaining elevated WUE, which is a crucial component of breeding programmes because of its yields in drought-prone areas. Any WUE is impacted by changes above the soil surface because they have an impact on the soil water balance by the evaporation and penetration of soil water. The majority of the chickpea crop is grown on residual moisture; however, additional irrigation can increase yields. At some sites in India, irrigation during the pre-flowering stage and at the beginning of the pod fill led to an increase in yield. Chickpeas’ reproductive cycle was prolonged by irrigation, which also increased plant biomass and increased the number of pods per plant.

The greatest sustained surface winds of tropical storms range from 39 to 73 mph, and they are fast rotating storm systems with an organized centre over warm tropical oceans. These storms have a wide range in size and can cause a variety of dangers for the impacted areas, including tornadoes, catastrophic winds, coastal floods, and inland flooding. The effects of tropical cyclones on drought have been extensively studied, but less research has been conducted on how smaller tropical storms affect the severity of drought. According to research, rainfall is not necessarily inversely correlated with the strength of a tropical cyclone; therefore, tropical storms can sometimes provide more rain than expected. The question of whether tropical storms can help to lessen and mitigate drought conditions is now being researched. Water deficit and surplus are related to drought and tropical storms (TS), respectively. When it comes to monitoring dryness, soil moisture is a crucial element of the hydrological cycle, since it reflects the water that TS rainfall has penetrated or stored. Soil moisture data can be used to determine whether TS can alleviate extremely severe drought situations [33]. The authors calculated the frequency of TS afflicted places in the US, including the ratio of droughts that TS exacerbated and alleviated, and the regions where TS have a significant impact on the offset of drought. Based on a high-resolution data set, the findings demonstrate extensive spatial information about the offset of drought conditions and offer potential guidance for future drought and TS mitigation.

Drought has a substantial influence on crop growth and photosynthesis, both of which are directly related to production [34,35]. Drought researchers must assess growth as well as physiological responses such as chlorophyll index [36], relative water content [37], membrane stability index, and biomass when determining the influence of drought on various crop metrics. The quantitative character of attributes and the prevalence of linkage between desired and undesired genes make developing drought-tolerant agricultural variants difficult [38]. Many experiments on the effects of drought on numerous chickpea features, such as root attributes, shoot biomass, and early maturity, have been conducted [39]. In this crop, various experiments have been performed successfully and published with specific conclusions on different aspects, such as morphological, physiological, biochemical, and molecular characteristics [40,41,42].

Advances genomics has made it possible to tag genes [43] associated to agronomic qualities, as well as the tolerance/resistance to abiotic and biotic challenges [44]. It is playing a significant role in the transfer of labelled genes through molecular breeding [45,46], quickly and accurately. In chickpeas, microsatellite and sequence-tagged microsatellite site markers have been found to be more beneficial [47,48]. As stress resistance/tolerance is governed by numerous genes [49], quantitative trait loci (QTL) mapping has proven to be effective in identifying and tagging the genes [50] involved for disease resistance/tolerance in plants. Foreground, recombinant and background selection are all examples of marker-assisted backcrossing. Linkage disequilibrium (LD) and association mapping are also determined using markers [51].

Next-generation sequencing (NGS) is a segment of revolutionary biology being a frontier area in crop science and produces correct data, with the results of significant throughput [52] and reduction in the need for fragment-cloning processes, which were the initial requirement for Sanger sequencing. NGS is used for the identification and mapping of mutations in targeted genotype [53,54]. Aside from whole genome sequencing (WGS), NGS also provides a platform for whole transcriptome shotgun sequencing, which is also termed as RNA sequencing (RNA-seq) [55,56] and whole-exome sequencing [57], which exhibits for functional variations [58], targeted or candidate gene sequencing [59].In the examination of large numbers of samples, RNA-seq enables a more precise and sensitive measurement of gene expression levels than microarrays [60].

Transcriptomics is the technology used to study the transcriptome of an organism [61]. Transcriptome is the complete set of genes [62] expressed under specific conditions by the genome of the targeted organism. MicroRNA (miRNA), transfer RNA (tRNA), messenger RNA (mRNA), ribosomal RNA (rRNA) and other non-coding RNA are all found in the transcriptome (ncRNA). Transcriptomics of chickpea [63] has provided insight into mechanisms of drought tolerance/avoidance, as well as pathogenesis-related and developmental processes [64]. Transcriptomics may undoubtedly have a greater impact on chickpea breeding in the future, including the use of microarrays.

Proteomics is the study of whole protein complement in a cell, tissue or organism in detail [65]. Mass spectrometry and protein microarrays can be used to analyse the proteome [66].

Role of various genes of plants under drought stress conditions have been recognized clearly [67]. Drought responsive mechanisms are activated in response to drought stress, which is a regular occurrence in plants. Morphological and structural changes [68], drought-resistant gene expression, hormonal and other biochemical changes are among these pathways [69]. Environmental stresses have the ability to change the developmental behaviour of plants. These alterations in plant growth and development [70,71] mostly resulted in lower yields [72]. We attempted to review the status and progress based on the existing literature on drought stress tests conducted in the chickpea.

## 2. Drought in Chickpea

Chickpea productivity has been found to be around 995 kg ha^−1^ on a global scale, which is quite low [73]. Drought, terminal heat [74], excessive salt, and cold are abiotic variables [75], whereas *Ascochyta* blight, *Fusarium* wilt, and *Helicoverpa* are biotic factors that have been recognised as key drivers of yield reduction in chickpea [76]. Drought stress was identified as a major cause in around 50% of chickpea output losses worldwide.

Several factors are responsible for complexity of drought stress (Table 1), including severity of drought, stage of crop, and duration of drought stress [77]. Two types of drought stresses, i.e., terminal and intermittent, have been reported with their impacts on crop plants [78]. During terminal drought, soil water availability diminishes over time, potentially leading to severe drought stress later in crop development. Intermittent drought is defined as a series of short episodes of insufficient rain or irrigation that occur at different times during the growing season [79]. Due to its limited cultivation on marginal terrain, chickpea is suffering from terminal drought stress. Intermittent and terminal drought stress is caused by breaks in rainfall combined with less moisture in terminal growth stages [80]. Apart from morpho-physiological factors various genes and proteins are also responsible for drought tolerance in chickpea crop (Table 2).

**Table 1 life-12-01846-t001:** Relevance of various physiological traits contributing to drought adaptation in chickpea.

Physiological Traits	Related with	References
Early phenology (early flowering, early podding)	Drought escape/conservative water-use strategy	[81,82,83]
Crop growth rate	High water harvest	[47]
Shoot biomass	High shoot biomass at maturity contribute to a higher grain yield under drought	[84]
Pod abortion and seed filling	High seed/grain yield could help in drought and heat stress tolerance	[85]
Biomass partitioning	Greater biomass partitioning to grain helps in drought and heat stress tolerance	[46,47,86]
Pod number; high pod number	Grain yield and contributes to heat, drought tolerance	[87]
Pod production	Number of pods/plants is more affected at early stage than late stage under drought stress	[88]
Specific leaf area	SLA has a positive effect on grain yield at reproductive stage	[89]
Cell membrane stability	Related to drought, heat, and cold tolerance	[30,90,91,92]
Canopy temperature depression	Cooler canopy contributes to drought avoidance and has a positive association with seed yield under drought stress, and it also contributes to heat stress tolerance	[93,94,95]
Canopy conductance	Associated to both heat and drought stress tolerance	[96]
Carbon isotope Discrimination	Transpiration efficiency	[97]
Recycling of CO_2_ inside the pod	Maintain seed filling	[98]
Antioxidants enzymes, proline, anthocyanin content, trehalose, sucrose, and nonreducing sugars	Increase in antioxidant enzymes, proline, trehalose and anthocyanin content during vegetative stage causes drought and cold stress tolerance	[99]
Relative water content	Increase in relative water content causes drought stress tolerance	[100,101]
Chlorophyll content; carotenoid content	Higher chlorophyll content and carotenoid content helps in heat stress tolerance	[55,96]
(Na+ and K+) ion uptake	(Na+ and K+) ion uptake cause drought tolerance	[102]
Chlorophyll a fluorescence FO, FM, PSII, ETR, FV/FM	Enable preventing PSII photochemistry from damage and helps in both drought and heat stress tolerance	[102,103]
Plant transpiration rate	Low plant transpiration rate helps in conserving soil water	[104,105]
Transpiration efficiency	It decides ultimate yield	[106,107]
Early vigour	Associated to both heat and drought stress tolerance	[108]
Pollen traits (pollen viability, fertility, and pollentube germination)	High pollen viability and fertility under heat stress are associated to heat stress tolerance	[109]
Abscisic acid (ABA)	Under drought increase in ABA causes closure of stomata, thus reducing assimilate production that leads to the inhibition of seed set	[108]
Root architectural trait prolific root system, root branch, root density root depth, root area, and root volume	Prolific root system is associated to grain yield	[47]
	Deep rooting helps in using conserved soil moisture from subsoil and helps in avoiding terminal drought stress	[108]

**Table 2 life-12-01846-t002:** List of some genes conferring adaptation to drought and other abiotic stresses in chickpea.

Treatment	Traits	Gene	References
Drought	Abiotic stress response	CarERF116	[110]
Drought	Biotic and abiotic stresses	Aquaporins gene family (*CaAQPs*)	[111]
Drought	Drought stress response	DEGs	[112]
Drought, heat and cold stress	Process of plant development	CarLEA4	[113]
Drought and heat stress	Root traits, plat morphology, transpiration, and yield traits	Marker–trait association	[47]

Due to the overabundance of wheat in irrigated areas in India, chickpea growth is primarily limited to rainfed areas. Crops in rainfed areas are experiencing water shortages, particularly during the sowing and terminal growth periods.

Soil and plant management are important for minimizing water stress. For this purpose, various experiments have been conducted with the applications of different agents. Gypsum can enhance overall plant growth, since it is a moderately soluble source of the crucial plant nutrients, calcium and sulphur. Gypsum supplements can also enhance the physical and chemical characteristics of soils, hence lowering nutrient concentrations in surface water runoff and reducing soil erosion losses. The most often used addition for reclaiming sodic soil is gypsum, which can also be found in synthetic soils used in nursery, greenhouse, and landscaping applications. Gypsum can be used for a variety of purposes in agriculture and horticulture, which could be advantageous to users. There are currently no recognized standards that outline the broad best management practices for using gypsum in agricultural applications.

Drought tolerance is a complex phenomenon that involves defence mechanisms as well as stress-induced signal responses [114,115]. Drought stress triggers a number of physiological, biochemical, and molecular responses (Figure 1) that can be classified into six categories: drought escape [116], avoidance [117], tolerance [118], resistance [119], abandonment [120], and drought adaptation [12,121]. Some chickpea genotypes have been identified as drought sensitive [122] and others as drought tolerant [123,124]. Plant breeders apply different ways of selection and development of drought tolerant crop genotypes. Different strategies are important to protect plants from harmful effects of drought.

Drought escape is the capacity of the plant to complete its life cycle before experiencing a major water deficit. Drought escape causes early flowering and maturity, as well as better yield potential, allowing plants to finish reproduction before drought strikes [125]. Crop longevity is governed in part by genotype and in part by the environment, and it impacts the crop’s ability to withstand climatic conditions such as drought. To achieve large seed yields, it is necessary to match the plant growth time to soil moisture availability. The genotypes with early maturity have the capacity to escape the terminal drought stress, whereas the genotypes with late maturity generally needs well-watered environments. The length of the growing phase and yield potential are positively associated with each other. In this consequence, the development of shorter duration crop is important for the reliable management of drought stress in the chickpea. The timing of flowering is a key feature of a plant’s response to extreme drought and high temperatures [126]. Early maturity lets the crop circumvent the passé of stress, hence short duration cultivars may be produced to minimise production loss from terminal dryness. However, under ideal growth conditions, the yield is often associated with crop duration, and any decrease in crop interval underneath the optimal will tax yield [29,52].

Drought escape is a critical strategy for preventing chickpea crop from drought [12]. Water supply is harmonised with phenological development in drought escape. Early maturity aids in escaping terminal dryness and is a key feature in germplasm screening. However, growers are frequently incapable to reorganize for early planting owing to climatic factors [52].

Drought avoidance is explained as a plant’s aptitude to retain a high tissue water potential contempt in a lack of soil moisture [125]. Processes involved in the enhancement of water intake, its storage in plant cells, and limiting water loss are associated with drought avoidance. Other mechanisms, including deep rooting, increased level of hydraulic conductance, reduced level of epidermal conductance, radiation absorption and reduced leaf area have also been reported to be linked with drought avoidance in plants. Deep rooting promotes water intake, which is helpful in reducing water losses. In the chickpea, the stomata remains closed during the day to minimise water loss during drought, and as a result, the carbon assimilation is impeded, lowering production [126,127].

Root biomass plays major role in absorbing water [128], as it is advantageous even in the condition of less moisture in the soil. It means there is a linkage between the root system and water stress tolerance [129,130], thus, in the current scenario, breeders are focused in the development of cultivars with larger root systems [131]. From integrating large root features, cultivars have been developed by chickpea breeders with increased drought tolerance [43,132]. Because root size is governed by intrinsic genetic variables [133,134] and modified by multiple environmental signals, such as nutrition and moisture accessibility in the soil, it is a complicated feature [135]. During the vegetative growth stage, susceptible genotypes absorb more water than tolerant genotypes, whereas tolerant genotypes absorb more water during the reproductive stage [136]. The intake of water during the vegetative as well as reproductive stages of plants has a direct relation with seed yield [137]. The importance of roots, rather than just root growth, is determined by their temporal water intake [138]. The best method for screening the germplasm for water usage competence (WUE) is carbon isotope discernment (13C), and this method has also been adopted in the chickpea [139,140].

One of the important impacts of drought stress is stomatal closure. Drought stress reduces the stomatal conductance and transpiration rate. This declines the CO_2_ fixation and photosynthesis due to the reduction in the internal CO_2_ concentration of the leaf (Ci). All of these factors have their role on the reduction in yield due to the reduced rate of photosynthesis [141].

The reduced rate of photosynthesis is directly related to extreme drought stress, and it is a result of the decreased chlorophyll content. Because of the lowered chlorophyll content, continuous poor moisture availability reduces light collecting capacity, triggering the generation of reactive oxygen species due to excessive energy absorption [142]. This is also a cause of damaged photosynthetic machinery. The principal cause of chlorophyll depletion is reactive oxygen species [143]. Reduction in photosynthetic activities under drought stress, have been experimented in chickpea genotypes [55] and this reduction was found to be linked to reduced ATP synthesis [144,145]. The yield reduction in chickpea genotypes due to the flower and pod drop under heat and drought stress circumstances was also noticed [146,147].

The leaf surface is also an imperative characteristic of plants in relation to drought stress. As tiny leaf surfaces lose less water [148], waxy leaves have high water preservation potential. Waxy leaves have the ability of a reflectance of irradiation and the reduction of water loss. This helps in the reduction of leaf temperature and provides tolerance against drought condition. The preservation of water in leaves with a reduced leaf temperature are directly related to the drought tolerant behaviour of plants. Drought stress raises leaf temperature in a variable manner, as tolerance genotypes have lower leaf temperatures than sensitive genotypes [149]. One drought tolerant chickpea variety ‘Gokce’ has been developed by ICARDA through the gene pyramiding method, which can be survived under severe drought conditions. This variety possess some other important features, such as early maturity, resistance to *Ascochyta* blight, increased seed size, and suitability for mechanised harvesting [150].

The drought tolerance refers to a plant’s ability to maintain its metabolism in a water shortage [35] condition with low tissue water potential [58]. Two types of traits are responsible for the drought tolerance in plants, i.e., constitutive characters and acquired behaviours. The constitutive traits affect the yield at mild to moderate levels of drought stress, whereas the acquired traits affect the yield at severe levels of drought stress. Drought tolerance features are largely concerned with cellular structural protection against the effects of cellular dehydration. Due to a reduction in the plant tissue water content, dehydrins and late embryogenesis of abundant (LEA) proteins accumulate [151]. These proteins act as chaperones [152].

In recent years, the role of reactive oxygen species (ROS) in stress signalling has been widely researched and evaluated [153,154]. The extreme creation of ROS causes oxidative damage and, lastly, cell death [155]. The role of ROS as a signalling molecule or in the oxidative damage depends upon the equilibrium between production and the scavenging of them [156]. The scavenging of ROS under drought stress depends upon the action of antioxidants in the cell [157,158].

Pushpavalli et al. [159] emphasised the need of selecting chickpea genotypes that can withstand various shocks rather than simply one. High temperature stress, in addition to drought stress, is a new threat to chickpea production [33,160]. According to Kalra et al. [161], a temperature increase of 18 °C above a particular threshold causes a significant loss in chickpea output. Furthermore, it is predicted that a global temperature increase of 2–38 °C, along with erratic rainfall patterns, would pose a threat to chickpea yield. In agriculture, the yield is the most important parameter for crops, and a reduction in yield cannot be compromised at any level. There is a strong association between drought tolerance with yield in a crop [122,162]. This is because yield-related traits of crops have been found to be sensitive under drought stress [163].

## 3. Antioxidant Defence

Plants have multi-level systems of antioxidant defence [124] with main function to maintain homeostasis inside the cell. This system counteracts ROS and protects the cell from oxidative damage. In the absence of the sufficient quantity of an antioxidant to neutralize ROS, reactions such as biomolecule oxidation, lipid peroxidation, and protein damage, as well as nucleic acid (DNA, RNA) oxidation and apoptosis activation, may occur [90].

The antioxidant defence system has both enzymatic and non-enzymatic components. The enzymatic component involves superoxide dismutase, catalase, peroxidase, ascorbate peroxidase, and glutathione reductase. However, the non-enzymatic component involves cysteine, reduced glutathione, and ascorbic acid [164,165].

In plants, ascorbate peroxidase is an important antioxidant enzyme, and glutathione reductase is important for sustaining the reduced glutathione pool during stress [166]. In various plants, two glutathione reductase corresponding to deoxyribonucleic acids have been recognized, one producing cytosolic isoforms and the other encoding glutathione reductase proteins, which dually embattle chloroplasts and mitochondria [127]. Superoxide dismutase is a key enzyme that catalyses the detachment of two superoxide molecules into O_2_ and H_2_O_2_ [167]. The drought tolerance of a particular plant species can be linked to increased antioxidant enzyme activity [168].

Proline appears to play a variety of activities under stress situations as a multifunctional amino acid, including stabilizing proteins, membranes, and subcellular structures as well as defending cellular functioning by scavenging reactive oxygen species (ROS). The functional diversification of proline metabolism is more complicated as a result of the compartmentalization of proline production and degradation in the cytosol, chloroplast, and mitochondria. When the electron transport chain is saturated under stressful circumstances, the increased rate of proline production in the chloroplast can help to stabilize the redox balance and maintain cellular homeostasis by dissipating the excess reducing potential. Proline is one of the most widely dispersed suitable solutes and a key component of plant stress resistance that increases in plants under adverse environmental conditions. Proline serves as a superb osmolyte and also has important functions as a metal chelator, antioxidant defence molecule, and stress signalling molecule. By regulating mitochondrial activity, affecting cell growth, inducing certain gene expression, and stabilising membranes, it promotes stress tolerance by reducing electrolyte leakage, bringing ROS concentrations back into normal levels, and promoting stress recovery.

One of the elements driving drought resistance in the chickpea is proline build up in different plant sections due to the increased activity of proline synthesising enzymes [169]. Drought tolerant genotypes of chickpea had higher proline contents than sensitive genotypes [130]. Earlier, an increase in the leaf proline concentration under water-deprived conditions indicates an efficient osmotic regulating system in the chickpea. To modify the osmotic potential, proline, glycine betaine, and soluble carbohydrates are accumulated in response to drought stress [170].

## 4. Plant Growth Regulators

Plant growth regulators, basically known as phytohormones, can be administered externally or synthesised inside the plant [171]. Auxins, cytokinins, gibberellins, ethylene, and abscisic acid have all been referred to as plant growth regulators. However, in recent studies, brassinosteroids (BRs) and various compounds of jasmonic acid, cytokinin, salicylic acid, strigolactones, and some peptides have been identified as plant hormones. The concentrations of Auxins, gibberellins, and cytokinin are negatively related to drought, but abscisic acid and ethylene have a positive association [172]. Drought stress inhibits the formation of endogenous auxins, which is frequently accompanied by a rise in the levels of abscisic acid and ethylene [173].

Abscisic acid and cytokinin are thought to play opposing functions in drought stress. Under water stress, an upsurge in abscisic acid and a diminution in cytokinin levels favour stomatal closure and minimise water loss by transpiration [174]. Abscisic acid affects the relative growth rates of different plant parts, such as the root-to-shoot dry weight ratio, leaf area development inhibition, and the generation of prolific and deeper roots. It can influence the rate of transpiration by closing the stomata, and it may be implicated in the machinery providing drought tolerance in plants. Under drought conditions, ABA formation inhibits the lateral root growth [175].

Ethylene is a growth inhibitory hormone that acts as a part in both inhibiting and stimulating growth in response to environmental factors. The plants can maximise growth and resist abiotic challenges such as drought to avoid this adversity, and this response also requires ethylene synthesis [134].

Plant growth and development are known to be affected by polyamines. There has been an increasing interest in the role of polyamines in the plant defence against environmental stressors, and substantial research energies have been conducted in the last twenty years. The overexpression of the apple spermidine synthase gene, for example, results in high amounts of spermidine synthase, which enhances abiotic stress resistance, for instance, drought tolerance [176].

## 5. Role of Conventional Breeding

Breeders commonly employ traditional breeding techniques such as introduction, selection, hybridization, and mutation [106]. Hybridization is used to blend the desired characteristics from different parents into a single cultivar [85]. Any hybridization program’s success hinges on the selection of proper parents. Single, multiple, and three-way crosses [107] have all been employed for the hybridisations in the chickpea crop [177]. Among the different branches of breeding technology, the mutation breeding has been found as a powerful strategy [178] for the creation of genetic variability in crops [179]. It is considered under advanced breeding technologies [180]. According to Kumar et al. [181], fifteen chickpea varieties have been developed through mutation breeding and most of them are under the cultivation chain. The first chickpea variety developed in the year 1984 through mutation breeding in India was Kiran (RSG-2), which was the mutant form of RSG-10. This variety possess higher numbers of pods, early maturity, high yield, and tolerance to salinity stress [182].

The evaluation of different genotypes of a plant species in response to the drought controlled condition is needed [183]. In different studies, the phenopsis has been used as a non-automated control-guided drought screening method [145,184,185] to examine the performance of several *Arabidopsis* ecotypes [186]. In this regard, the *ERECTA* gene [147] responsible for the growth and development of the plant as well as the stomatal development, the *ESKIMO1* gene [187], governs the plant water relation in *Arabidopsis,* which have been examined well. Similarly, some biosynthesis genes in the chickpea [188] have been well studied using controlled drought.

New cultivars, landraces, wild relatives, or a new crop species for the region could all be introduced. By using this technique, it is feasible to find a desirable genotype with a higher yield and better environmental tolerance, while also increasing the genetic variety. Through the international exchange of the germplasm and the inclusion of the crop of wild relatives and landraces, significant progress has been made in recent decades in enhancing the genetic diversity of the cultivated chickpea. Landraces are an important resource of novel genes in crop breeding. Landraces may possess genes for resistance against various biotic as well as abiotic stresses. For the identification of drought tolerant chickpea landraces, a field study was conducted by Kumar et al. [189]. The experiment included 37 chickpea landraces collected from ICARDA. Based on various morphological as well as physiological parameters, two landraces viz.,IG5856 (Jordan) and IG5904 (Iraq), were identified as drought tolerant.

## 6. Role of Molecular Breeding

### Genetic Diversity

Complex abiotic stress such as drought requires a large group of genetic resources [190] to study the genetics of these stresses authentically. To fulfil the objective of molecular breeding in crop improvement, it is important to characterize the plant genetic resources. In molecular breeding, the characterization of plant genetic resources depends on the availability of DNA-based markers [191]. The molecular markers may be hybridization-based or PCR-based, depending on the technique of the detection of nucleotide variation [192]. The restriction fragment length of polymorphism (RFLP)is one of the molecular markers based on hybridization. Random-amplified polymorphic DNA (RAPD), amplified fragment length polymorphism (AFLP), simple sequence repeats (SSR) or microsatellite, sequence-tagged sites (STS), and cleaved amplified polymorphic sequence (CAPS) are all PCR-dependent molecular markers [158]. In the advanced category, single nucleotide polymorphism (SNP), single feature polymorphism (SFP), and diversity array technology (DArT) have been included [193]. In the field of crop improvement, molecular markers are categorized into dominant and co-dominant. The multi-locus markers (RAPD, ISSR, AFLP, etc.) come under the dominant category, while the single locus markers (SSR, STS, etc.) come under the co-dominant. This categorization of markers is basically based on their efficiency to discriminate homozygous and heterozygous genotypes. Dominant markers cannot differentiate homozygous and heterozygous genotypes. However, co-dominant markers have the ability to differentiate them.

In comparison to molecular markers, i.e., the hybridisation-based RFLP, the PCR-based RAPD and ISSR, SSR, and biochemical markers, i.e., isozyme, have a low polymorphic ability, which could be related to the decreased polymorphism in structural genes in the chickpea genome [194].

In some of the previous studies, the RAPD markers were utilised to detect genetic relationships amongst *Cicer* species [195]. The non-reproducibility nature of this dominant character is the main reason of the limited applicability of it [196]. However, the sequence characterised the amplified region (SCAR) markers developed with the use of RAPD markers, which are more suitable for the detection of the desired gene in crops, including the chickpea [197]. Amplification fragment length polymorphisms (AFLPs) were also reported to be uncommon in *Cicer arietinum* [198]. However, the availability of some reports on use of these markers in chickpea for diversity analysis [199] and the screening of abiotic as well as biotic resistant genotypes [200] proved their importance.

Microsatellite and STMS (sequence-tagged microsatellite site) markers are numerous, scattered throughout the genome, and highly polymorphic [198]. STMS raises the likelihood of finding polymorphism by a factor of ten. As a result, any genetic enhancement initiative should begin with a study of genetic variability. These markers become an important part of molecular breeding in the chickpea [201].

More than 3000 microsatellites [202,203], 15,000 DArT arrays [204,205], and SNPs [206] markers have been developed in the last few years in the chickpea. Because of a few of the specific characteristics, including co-dominance, abundance, repeatability, higher polymorphism and large genome coverage, SSR markers have proven their efficiency in the field of molecular breeding [207]. Subsequently, the data obtained after the use of SSR markers for molecular characterization or fingerprinting can be used to determine the genotypic identity of an individual. The applications of ISSR markers for the genotypic identification of chickpea genotypes in association with the seed germination and flowering time reported recently by Yadav et al. [208]. In this sequence, SNPs/InDels were also used recently for the gene identification and analysis in chickpea [209]. Basu et al. [210] identified SNPs linked with seed yield and Rajkumar et al. [211] identified SNPs linked with seed size and seed weight in the chickpea.

## 7. QTLs and Their Relevance with Drought Tolerance in Chickpea

A crucial requirement for identifying and integrating genes in linkage maps for marker–aided selection (MAS) is the knowledge of the agronomic trait inheritance [212]. The marker-assisted selection [54] and mapping of QTL (Quantitative Trait Loci) have been proposed to improve chickpea productivity [213]. Linkage map construction [214] and attribute mapping [215] were both conducted with available markers in the chickpea. Many research groups have focused their studies on abiotic stresses [216,217]. After completion of the sequencing of the desi and kabuli chickpea genomes [218], a genome-wide physical map was also generated. Furthermore, QTL studies have also been carried out in the chickpea (Table 3) to better understand the genetics of drought tolerance [217,219] and salt tolerance [220]. Varshney et al. [221] identified ‘QTL-hotspot’ regions that contain QTLs for a number of drought-related characteristics in the chickpea. They also reported the linkage between QTL hotspots and SSR markers. In the marker-assisted selection, the chickpea genotype ICC-4958 is used as a control for root studies under drought condition due to the large root character. This character makes this genotype suitable to use as a parent for transferring drought tolerance QTL-hotspot regions into the desired genotype. Recently, Muriuki et al. [222] also evaluated the root traits of some chickpea genotypes under drought stress and found that some of the *desi* genotypes (ICC4958, ICCV 00108, ICCV 92944 and ICCV 92318) performed fine.

**Table 3 life-12-01846-t003:** List of QTLs identified for drought tolerance in chickpea.

Mapping Approach	Numbers of QTLs	Markers Used	Statistical Method Used	References
Biparental	15 QTLs	SSR		[151]
Biparental	93 QTLs	SSR	Composite interval mapping-epistatic mapping (ICIM-EPI)	[213]
Biparental and backcross	QTL-hotspot	SSR, AFLP		[13]
Biparental	QTL-hotspot	SSR	Composite interval Mapping	[214]
GWAS	312 significant model MTAs	DArT, SNP	Mixed linear	[58]
Biparental	164 main-effect QTLs	SNP, CAPS	Composite interval mapping	[215]
Biparental	QTL-hotspot_ a(15genes)	SNP	ICIM-ADD mapping method	[216]
Biparental	3 candidates Genes	SNP		[217]
Biparental	12 QTLs	SNP		[218]
Biparental	21 QTLs	SNP	Composite interval mapping	[223]
GWAS	Several MTAs	SNP		[224,225]

The advanced genomics involves a genome-wide association study (GWAS) that helps researchers in the screening of a wide range of genotypes [226,227] with different phenotypic or agronomic characters, and it also helps in the identification of the variability present among them. GWAS also helps in the identification of the association between the marker and a specific trait of interest [28,228]. The majority of these connotation investigations used either GWAS or candidate gene sequencing. Recently, the GWAS-based association mapping for the drought tolerance in the chickpea and for salinity tolerance have been performed. Apart from these, other examples of the association mapping in the chickpea are also available, i.e., for iron and zinc concentration in seeds [229] and *Fusarium* wilt resistance [230,231].

In some of the earlier studies, a combined analysis of the GWAS and sequencing of candidate gene [223] has been found to be more suitable in crop improvement. The GWAS study on two sets of chickpea genotypes with a different degree of their response in drought conditions helped in the discrimination of these genotypes on the basis of the single nucleotide polymorphisms generated.

Scientific efforts made on the improvement of the chickpea crop made it possible to generate not only a bi-parental plant population but also multi-parent populations [224]. The need of a multi-parental population was due to issues such as narrow genetic variability and limited efficiency of the bi-parental population [225] during the multiple trait analysis [232]. Multi-parent advanced generation inter-cross (MAGIC) populations for the chickpea are being established [233] to create diverse patterns of recombination [234]. The purpose of creating multi-parent populations is to advance the precision of QTL mapping [235] and discover specific loci regulating to the trait of interest [236]. ICRISAT and ICARDA played a major role in the development of a few MAGIC populations in the chickpea. One example of the MAGIC population developed at ICRISAT is the results of crossing eight varieties and advance breeding lines (ICC 4958, ICCV 10, JAKI 9218, JG 11, JG 130, JG 16, ICCV 97105, and ICCV 00108) with eight different founder parents [6,237]. Similar to this one, the MAGIC population developed at ICARDA was the result of crossing 12 different parents [238]. These plant populations accelerate the detection, isolation, and transfer of critical candidate genes to help in the development of chickpea varieties with superior agronomic traits [237]. One more approach (target-induced local lesions in genome -TILLING) [239] was adopted in the validation of the drought responsive gene in chickpea [58]. The selection and further use of agronomically superior genotypes of chickpea for the development of new varieties are the basic objectives of breeders involved in the chickpea crop improvement [240].

## 8. Attempts to Develop Drought Tolerant Varieties

One of the drought tolerant high yielding Ethiopian chickpea varieties,‘Geletu’, was developed and released in the year 2019 through the marker-assisted back-crossing after multi-location trials. During the development of this variety, the‘QTL-hotspot’ linked to drought tolerance was introgressed into an Indian chickpea cultivar JG11 from ICC4958 (gldc.cgiar.org). Recurrent selection is a crucial breeding technique used to increase crop plant populations. It is a productive method used in plant breeding to enhance the quantitative traits through repeated crossing and selection. Among the genomics-assisted selection methods, the marker-assisted back crossing has been found better for the introgression of the targeted region of the genome into a desired genotype. Consequently, the introgression of the ‘QTL hotspot’ region for the development of the drought-tolerant chickpea through molecular breeding has been found effective. Similar to this, numerous drought tolerance characters were introgressed into three elite Indian chickpea varieties: Pusa 372, Pusa 362, and DCP 92-3from ICC 4958. Recently, drought tolerant root traits have been introgressed into Kenyan chickpea varieties using the marker-assisted backcrossing approach [241].

Initially, the Pusa 372, chickpea variety was released as a drought tolerant variety for cultivation in the central, north-east, and north-west plains zones. However, under drought conditions, this variety’s output has decreased in recent years. To enhance drought tolerance in this variety, the MABC approach was adopted to introgress the ‘QTL-hotspot’ region from ICC 4958 into ‘Pusa 372′. Recently, the Pusa 372 was released with improved drought tolerance under the name ‘Pusa 10216′. This improved chickpea variety is the example of the first enhanced drought tolerant chickpea variety developed through the MABC approach. ICRISAT in collaboration with other research institutes in India is in the process of the development and release of drought tolerant chickpea varieties, i.e., IPC L4-14 and BGM 4005.Both of these varieties were developed by transferring a ‘QTL-hotspot’ from ICC4958 into DCP92-3 and Pusa362, respectively (https://www.icrisat.org/new-climate-resilient-disease-resistant-chickpea-varieties-coming-farmers-way/, accessed on 20 October 2022).

## 9. Whole-Genome Re-Sequencing

Whole-genome sequencing is the most thorough NGS technology [242], allowing for the complete genome sequencing and identification of variations in both exonic and non-coding areas, as well as the structural variant detection. Due to a paucity of genetic knowledge, the chickpea was formerly referred to as an orphan crop well adopted to suboptimal growing environments [243] However, researchers published the first draft genomes of the desi and kabuli chickpeas in 2013. The chickpea genome sequencing was based on advances in high-throughput sequencing and next-generation approaches. The BAC end genetic map and DArT markers were used to offer information on SSR and SNP molecular markers [244]. Both the kabuli and desi chickpea genomes have been updated, as well as a comparative examination of the two varieties. The QTLs associated to drought tolerance were reported by Jaganathan et al. [243]. The drought-responsive genomic areas were identified and employed in breeding approaches such as the marker-assisted gene interrogation and genetic gain to improve production in harsh climatic circumstances.

After publication of the draft genome sequence of the chickpea, the sequencing-based technique for the improvement of this crop has open multiple windows [244]. Furthermore, re-sequencing of a large number of chickpea assents collected from 45 nations enabled the identification of various candidate genes with their associations to a large number of agronomical characters [245]. The results of these experiments revealed the origin and migration routes of chickpea in the world. Re-sequencing data helped in the identification of 50,590 SNPs, and this data was used to develop the ‘Axiom^®^CicerSNP Array’ [246]. This SNP platform is being employed in the character mapping and identification of QTLs. Recently, Rajkumar et al. [247] reported the re-sequencing of large and small seed chickpea genotypes and 266 SNPs associated with seed size and seed weight. The findings of the study may help in the selection and categorization of chickpea genotypes on the basis of the size and weight of their seeds.

Next generation sequencing technology [248] has made possible the development of new markers for the improvement of the chickpea [249]. One of the important concepts, ‘The 3000 Chickpea Genome Sequencing Initiative’ [250], is an important step in the field of chickpea improvement. This initiative is helpful in the identification of variations in genomic sequences (rare alleles, markers) and their role in the determination of various agronomic characteristics, including yield and resistance/tolerance, against biotic and abiotic stresses. A thorough map of variation in 3171 farmed and 195 wild accessions was recently published in a project to give publicly accessible tools for chickpea genomics research and breeding [250]. This study also demonstrated the variations among the cultivated and wild progenitors of chickpea.

## 10. Pangenome and Super-Pangenome

The availability of the whole genome sequences of multiple individuals makes it feasible to compare them for the identification of diversity among them [251]. This approach may be termed as a comparative genomics [252] analysis, which allows for the identification of bio-markers linked with taxonomic as well as morphological and functional characteristics [253]. In this sequence, the pangenome concept was arisen, which allows for the accurate and efficient comparison of the genomes of a wide range of individuals [254]. A recently proposed revolutionary approach known as super-pangenome, allows the construction of the pangenomes of many species within a specified genus [255]. These concepts facilitate the identification of novel variations among individuals from different sources. These advanced technologies have their importance in crop breeding due to their accuracy and efficiency. The construction of pangenomes has their advantages in the identification of signature genomic areas relevant to crop domestication and evolution. Chickpea landraces and varieties have been sequenced to build the pangenome. These pangenome may be coupled with phenotypic traits and alleles associated with various characteristics and may also help in the identification of abiotic stress tolerance [256]. Pangenomes also provide a platform for the accurate identification of target genes for genome editing using CRISPR-clustered, regularly interspaced, short palindromic repeat technology [257].

## 11. Omics Approaches

Complex genetic traits, including drought tolerance, need advanced tools for their dissection, along with crop improvement [258]. Multiple omics approaches have revolutionized the identification of genes [259] as well as the metabolic database [260].Investigations have been carried out on transcriptomic analysis with the applications of the NGS technology in chickpea [261]. Multiple examples on transcriptomics’ evaluation in the chickpea are available as developing seeds [262], development and function [263], tissue specificity [264], and salinity tolerance [265] as well as root transcriptomics for drought tolerance. About 20,162 ESTs in the chickpea under salt and drought stress circumstances have been reported (Table 4). Recently, Kaashyap et al. [266] performed a comparative flower transcriptomic analysis to analyse the reproductive success under the salinity stress in the chickpea.

**Table 4 life-12-01846-t004:** Advanced technologies adopted to identify drought responsive differentially expressed genes/ESTs in chickpea.

Differentially Expressed Genes/ESTs	Technique/Platform Used	References
1562 genes, 2592 genes	Illumina HiSeq 3000	[267]
1624 differentially expressed genes	Illumina platform	[103]
20,162 ESTs	-	[266]
53 ESTs	cDNA library	[268]
3062 unigenes	Suppression subtraction hybridization	[258]
44,639 differentially expressed sequences	Roche/454 and Illumina/Solexa	[269,270]
7532 unitags and 880 unitags	SuperSAGE	[267]
4053 and 1330	Illumina HiSeq 2000 platform	[271,272]
261 (shoot) and 169 (root)	Illumina TrueSeq RNA	[273]
15,947 differentially expressed genes	Illumina HiSeq 2000	[274]

The RNA-Seq technique has also been used to analyse differential regulation of genes under drought stress in the chickpea [112]. Kumar et al. [113] analysed the gene expression of polyethylene glycol-stimulated drought stress in the chickpea, and thousands of differentially expressed genes (DEGs) were identified. Earlier, DEGs were detected in the kabuli chickpea under drought conditions [275]. Using RNA-Seq technique, it was used for the development of an inclusive *C. arietinum* Gene Expression Atlas (CaGEA) based on a drought tolerant ICC 4958 cultivar [269]. The findings of this study also validated the ‘QTL hotspot’ for drought tolerance in the chickpea.

The regulation of metabolic activities plays a major role in maintaining the osmotic potential of the cell under drought stress [276]. With the applications of the metabolomics method, many important metabolites were identified with a different regulation pattern during a drought [277] as well as in the salinity [278] in the chickpea. Similar to this, in an earlier investigation conducted on *Arabidopsis thaliana*, various genes were identified with their similar contributions under both salinity and drought stresses. The production of similar metabolites under both abiotic stresses indicates a common tolerance mechanism for drought as well as in the salinity in plants.

The proteomics’ analysis has also been performed in the chickpea for the identification of changes at a protein level under abiotic stresses. Earlier, a comparative proteomics analysis was conducted on one chickpea cultivar (JG-62), and novel dehydration-responsive proteins were detected [279]. In this sequence, Jaiswal et al. [280] reported the role of Sad1/UNC-84 in dehydration signalling. Recently, Vessal et al. [281] analysed the proteomic responses of drought sensitive and tolerant chickpea cultivars and identified changes in terms of the requirement of relative leaf water content for tolerant and susceptible cultivars. Drought responsive root proteins were also analysed recently by Gupta et al. [282].

Phenomics is an emerging tool in plant research, and is used to describe the use of genomics in phenotyping [283]. Phenomics studies for different phenotypic traits as well as seed yield have been conducted for the drought tolerance in chickpea [32]. The drought tolerance in chickpea is determined through phenotyping, and for this purpose, high-throughput screening technologies [32,44,284] have been adopted.

## 12. Role of Candidate Genes

A contender or candidate gene is thought to be linked to a specific disease or phenotypic character [285], and the biological function(s) of that has been derived either directly or indirectly from other investigations, including, for instance, the genome-wide association studies [286], the traditional map-based positional cloning technique, and the more recent next-generation sequencing (NGS) method [287]. Candidate gene studies are low-cost and rapid to conduct, and they focus on finding genes that have already been linked to the disease and hence have a prior knowledge of gene function [288]. Few of the important candidate genes detected in chickpea for the abiotic stress tolerance are Snf-1-related kinase (*AKIN*), *DREB2A*, dehydrin (*DHN*), *CAP2*, and *Myb* transcription factor (MYB) [289], Table 5. Despite the fact that multiple genes have been linked to drought resistance, the association study based on candidate gene sequencing has received little attention.

**Table 5 life-12-01846-t005:** List of various genes/transcription factors and their roles in response to drought and other abiotic stresses in chickpea.

S.No.	Gene/Transcription Factor	Function	References
1	DREB	Dehydration responsive element binding proteins	[290]
2	Dehydrin (DHN)	Response to water stress	[291,292]
3	STPK	Drought stress	[293]
4	CAD	Response to abiotic stress	[294]
5	AMADH	Wound healing, abiotic stress responsive	[295,296]
6	TCS	Abiotic stresses tolerance	[297]
7	EREBP	Ethylene responsive	[298]
8	LEA Gene	Response to water stress	[299]
9	AKIN	Positive regulator of drought tolerance	[300]
10	Myb transcription factor	Stress	[301]
11	ASR	Abscisic acid stress and ripening gene	[302]
12	SuSy	Sucrose synthase	[303]
13	CAP2	Promoter of DREB2A	[297]
14	ERECTA	Transpiration efficiency regulator	[298]
15	SPS	Sucrose phosphate synthase	[300]
16	CAMTA	Salinity and drought tolerance	[304]
17	CarNAC4	Salt and Drought tolerance	[305]
18	CaNAC	Drought tolerance	[306]
19	CarERF	Drought stress	[109]
20	CaSWEET	Abiotic stress tolerance	[307]

## 13. Transcription Factors and Their Role in Drought Tolerance in Chickpea

Transcription factors induce the cis-elements in the promoter provinces of different stress-responsive genes to accelerate the countenance of various downstream genes, which have their part in the stress tolerance [308] of plants. According to Riechmann et al. [309], nearly 1500 transcription factors have been reported in the *Arabidopsis thaliana* genome, which have their part in stress-responsive gene expression.

## 14. Dehydration Responsive Element Binding Proteins (*DREB*s)

*DREBs* (dehydration responsive element binding proteins) are key plant transcription factors [12]. They are responsible for the regulation of many stresses’ responsive genes. One of the transcription factors, AtDREB1a, was identified from *Arabidopsis thaliana* [310], with its role under abiotic stress. Recently, Das et al. [290] reported a better performance of *AtDREB1a* transgenic chickpea lines under water stress conditions.

## 15. Dehydrin (*DHN*)

Dehydrin (*DHN*) are stress-responsive proteins that are found when the temperature is low or when the body is dehydrated [311]. Protein dehydrin protects the embryo and seed tissues under water scarcity [312]. A better performance of transgenic plants overexpressing *DHN* than wild-type plants [313] have been identified. The role of the *DHN* gene in the Pusa1103 and Pusa362 genotypes of the chickpea has been found to be linked with the drought tolerance. Furthermore, in comparison to other genotypes, these genotypes were recognised as drought tolerant due to their better response [291].

## 16. Serine/Threonine Protein Kinase (*STPK*) Gene

Serine/threonine protein kinase (*STPK*), tyrosine protein kinase (*TPK*), and histidine protein kinase (*HPK*) are the three types of eukaryotic protein kinases [314]. In Arabidopsis, chickpea, and rice, the *STPK* family gene *AtSnRK2.8* is reported with an enhanced degree of drought tolerance [293].

## 17. Cinnamyl Alcohol Dehydrogenase (*CAD*)

*CAD* is thought to be important in plant defence against a variety of biotic and abiotic stressors. When employing primers developed for the contig exhibiting match with the *CAD* gene of *Arabidopsis thaliana*, a homologue form of this gene was recovered from eight chickpea genotypes [294].

## 18. Ethylene-Responsive Element Binding Protein (*EREBP*) Gene

Ethylene-responsive element binding factors (ERFs) are a new type of transcription factor that are only found in plants. The ERF domain, a highly conserved DNA binding domain, is the protein family’s distinguishing trait. Primers for the chickpea were constructed using a contig sequence that was found to be identical to the *Arabidopsis thaliana* ethylene-responsive transcription factor. The amplification of eight chickpea genotypes yielded amplicons of around 400 bp. In plants, the *AP2*/*EREBP* genes have a variety of roles in developmental processes and stress responses [315].

## 19. Amino-Aldehyde Dehydrogenase (*AMADH*)

In some crops, an *AMADH* gene has been reported with its association to osmotic stress tolerance [316] by detoxifying hazardous aminoaldehydes; this gene has a function in physiological as well as metabolic responses under abiotic stresses [295]. On the basis of functional characterisation of the *AMADH* gene in *Arabidopsis* [317], the role of this gene should be examined in the chickpea.

## 20. ERECTA Gene

The *ERECTA* gene has a part in leaf organogenesis, lowering the density of the stomata on the leaf underside and thereby lowering evapotranspiration. It can also control the transpiration by the alteration of the leaf epidermal cell expansion, proliferation of mesophyll cells, and cell–cell interactions. The *ERECTA* gene has been demonstrated to regulate the growth and development of the organ and flower in *Arabidopsis* via encouraging cell proliferation [318]. Complementation tests on the wilting mutant *Arabidopsis* plants confirmed the involvement of the *ERECTA* gene to the water usage competence [319]. Pioneer Hi-Bred International, Inc. has patented the *ZmERECTA* genes from maize, which were implicated in the drought tolerance in crop plants.

## 21. Late-Embryogenesis Abundant (LEA) Proteins

The attainment of dehydration tolerance and the behaviour of plants to drought have been linked to late-embryogenesis abundant (LEA) proteins. Increased LEA and Dehydrin expression in genotypes during the vegetative, flowering, and podding stages could represent an adaptation to assist the plant survival by supplying the energy for growth and survival [29]. Leaf age inhibited arLEA4 expression, which changed during seed and pod development, including during germination. Drought, salt, heat, cold, ABA, IAA, GA_3_, and MeJA all significantly increased the *CarLEA4* expression. *CarLEA4* is a LEA assembly 4 protein that may participate in a variety of plant developmental processes as well as abiotic stress responses [299].

## 22. Myeloblastosis (MYB) Gene

Plants have a big transcription factor (TF) family, called the myeloblastosis (MYB) gene [320]. It plays a role in the secondary metabolism regulation, hormonal and climatic condition response, cell differentiation, and resistance to drought and other abiotic stimuli. Under drought stress, arrays of MYB-transcription factors are involved in the generation of epicuticular waxes [321]. These waxes seal the plant’s aerial component and reduce water loss through the leaf surfaces [322]. In an experiment, the root tissue of ICC 4958 (drought tolerant), ICC 1882 (drought sensitive), JG 11 (elite), and JG 11+ (introgression line) were employed to recognize the role of the 1R-MYB gene in the machinery of the drought tolerance in the chickpea. The findings of this experiment were suggested to conduct more experiments on this aspect in the chickpea. Recently, Caballo et al. [301] observed that CaRAX1/2a codes a MYB transcription factor that is exactly articulated in the meristem of chickpea. These results disclosed that the single flower gene (*SFL*) encodes for MYB, which works as a central factor responsible for the regulation of the numbers of flowers in chickpea inflorescence.

## 23. *S-Adenosylmethionine Synthetase* Gene

S-adenosylmethionine (*SAM*) is a precursor in the production of polyamines and ethylene [323]. In plants, the action of 1- aminocyclopropane-1-carboxylate (ACC) synthase and ACO (ACC oxidase) is responsible for ethylene biosynthesis, while the activity of *SAM* decarboxylase is responsible for spermidine and spermine production. The exogenous polyamine administration or overexpression of polyamine production genes has been demonstrated to improve abiotic stress tolerance. Primers were developed using a contig sequence that was comparable to the S-adenosylmethionine synthetase 1 (*SAM1*) gene of *Arabidopsis thaliana* for the isolation of the S-adenosylmethionine synthetase 1 gene homologue in the chickpea. The PCR amplification revealed amplicons of roughly 300 bp in eight chickpea genotypes [324].

Expression of *SAM* gene in pigeon pea (*Cajanus cajan* L.) was evaluated under drought, heavy metal (CdCl_2_), and cold stresses. The enhanced up-regulation of *SAM* gene in the leaves were recorded after three days [325].

## 24. Abscisic Acid Stress and Ripening Gene

Among many other genes, the abscisic acid stress and ripening (*ASR*) gene plays a critical role in controlling various plant stresses. The *ASR* gene has been reported in plants, and is induced by abscisic acid and different abiotic stresses during the process of fruit ripening [268]. Reports on *ASR* genes with their responses in different plant species under drought, salt, and cold stresses [326] confirm their role. Transgenic *Arabidopsis* demonstrated the over-expression of the *ASR* gene in response to drought and salt stresses [302]. Genotypes of rice also presented the association of the *ASR* gene expression [327]. Similarly, Cortés et al. [328] reported the potential significance of the *ASR1* gene in the common bean. In a recent study conducted on the chickpea under drought stress, increased *ASR* gene expression was observed. The increased expression may have helped the drought-tolerant chickpea genotypes function better under stress. This hypothetical *ASR* protein could have boosted the activity of the *ASR* gene as a transcription factor mediating drought responses in chickpeas.

## 25. ABRE-Binding Protein (*AREB*)

Various genes that are activated by abscisic acid (ABA) have been discovered to be drought stress-inducible. Such ABA-regulated genes have conserved cis-elements in their promoter regions known as ABA responsive elements (*ABREs*), which use bZIP-type *AREB/ABF* transcription factors to regulate the gene expression. ABA and water stress upregulate the expression of the *AREB/ABF* gene. Expression of *AREB* gene under drought stress has been reported by Yoshida et al. [329] in *Arabidopsis thanliana.*

## 26. Sucrose Synthase (*SuSy*) Gene

Sucrose synthase (*SuSy*) is a crucial enzyme that hydrolyzes sucrose directly to provide substrates for plant metabolism. It is also used as a bio-marker for plant sink strength [330]. Plant sink strength improvement could contribute to increased plant growth and yield [331]). In an experiment, cultivars and treatments had a strong and positive association between the seed dry weight at maturity and peak sucrose synthase movement. Sucrose synthase is a decent physiological indication to employ in chickpea breeding for larger seeds.

Sucrose synthase activity has a major role in chickpea seed growth. The supremacy of the sink, as measured by the sucrose synthase movement most of the time, hinges upon genetic features of a genotype along with the accessibility of water obtainable at seed filling [332]. In both the large-seeded kabuli and the small-seeded desi varieties, the water shortage reduced the enzyme action and seed size, but the higher enzyme action in the large-seeded kabuli, pre-dominantly at the late seed filling stage, seemed to persuade a better remobilization of the integrates from the pod wall and seed coat. The cotyledons’ greater sucrose synthase action is taken into account. The strong association between sucrose synthase activity during rapid seed filling and final seed dry weight accumulation, and therefore seed size, advises that the sink strength is an important element of the seed size in chickpea. The tight link between the sucrose synthase activity during rapid seed filling and final seed dry weight build up, and thus seed size, implies that the sink strength is a key factor in the chickpea seed growth. Higher cotyledon sucrose synthase activity is vital in breeding for better seed size in chickpeas, regardless of the growth environment [303].

## 27. *CAP2* Gene

The AP_2_ subgroup of proteins has two copies of the DNA-binding domain (BD), detached by an insertion province [333]. *CAP_2_* is a C-Repeat binding factor (CBF) that muddles to the DRE/CRT (dehydration responsive element/C-repeat element)(CCGAC) found in the promoters of abiotic-stress responsive genes. Dehydration, excessive salinity, and exogenous ABA treatment all enhanced *CAP_2_* gene expression. The incidence of roughly 60-amino-acid long AP_2_/ERF DNA-binding realms in these transcription regulators allows them to connect directly with GC-rich cis-acting elements (GCC box/C-repeat) in the promoter of their target genes. Ectopic expression of CAP_2_ in tobacco resulted in increased drought, salinity, and heat tolerance, as well as improved transgenic plant growth [334]. The enhanced accumulation of the CaZF transcript was caused by the transient expression of *CAP_2_* in chickpea leaves. *CAP_2_* activates the CaZF promoter through interacting with C-repeat elements (CRTs) in CaZF promoter, according to the gel mobility shift and transient promoter-reporter tests. The *CAP_2_* protein interacts with the CaZF promoter in vivo, according to a chromatin immunoprecipitation (ChIP) test [335].

## 28. Sucrose Phosphate Synthase (*SPS*) Gene

The sucrose phosphate synthase (*SPS*) gene has an important function in the sucrose production in different plant species. It regulates sucrose metabolism in drought sensitive and tolerant genotypes [300]. Sucrose biosynthesis and sucrose degradation determines the level of sucrose in a genotype, and an optimum level of it is important for growth and development under environmental stress in plants [336]. Some reports are available on the unchanged or decreased level of the *SPS* activity in maize, potato, soybean, and some other crops, however, some reports advocate the increment in the *SPS* activity in rice, wheat crops, and *Arabidopsis* [336]. The significance of the *SPS* gene has been studied in chickpea under a low temperature by Sharma et al. [300].

## 29. Genome Editing Options

The genome editing approach, specially CRISPR-Cas9, has proved their efficiency in the development of climate resilient cultivars of different crops [337]. Two genes, *namely* RVE7 and 4CL, have been identified in the chickpea and their association with drought tolerance.The CRISPR/Cas9-mediated editing of the chickpea protoplast was reported for the first time by Badhan et al. [338], where they reported knock-outs of 4CL andRVE7 genes, which are associated with drought tolerance mechanisms. This report laid down a foundation for future genome editing options in the chickpea [339]. Genome editing approaches with the applications of CRISPR-Cas9 (Figure 2) may be helpful in the development of abiotic stress tolerance in chickpea genotypes including drought.

## 30. Conclusions

As previously stated, the changes in the plant shape and internal biochemical characteristics during drought stress have been extensively characterised in previous studies. Plant drought stress techniques can help us better use scientific means to improve plant tolerance to water shortage environments and increase crop yields, allowing us to play a larger role. As a result, by thoroughly examining and summarising the mechanisms of the chickpea plant response to drought, this study provides essential background knowledge and theoretical framework for selective breeding, cross breeding, and molecular breeding of the chickpea in the future. Drought tolerant land races/germplasm lines may be employed in classical as well as molecular breeding programmes to breed drought tolerant cultivars in future by using the available scientific data.

## Figures and Tables

**Figure 1 life-12-01846-f001:**
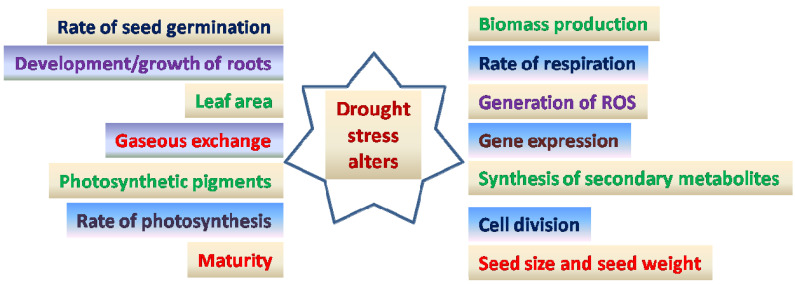
Diagrammatic representation of effects of drought stress on chickpea.

**Figure 2 life-12-01846-f002:**
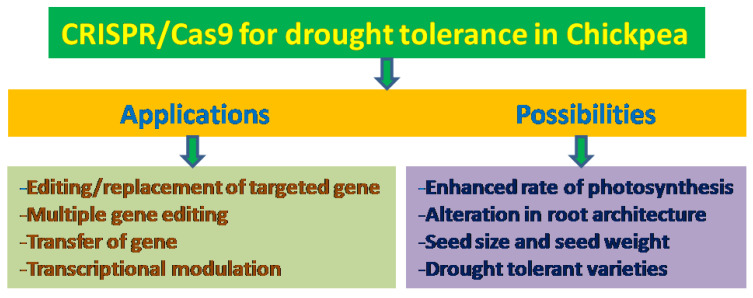
Diagrammatic representation of scopes of CRISPR/Cas9 applications and possibilities in chickpea improvement.

## Data Availability

Not applicable.

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
