# Peer review of "Molecular Breeding and Drought Tolerance in Chickpea"

_life, 2022, doi:10.3390/life12111846_

Round 1

Reviewer 1 Report

Comments

1) Comprehensive and well-done bibliographic review regarding mechanisms, genetics and improvement linked to drought tolerance (DT) that focuses on rainfed planting. However, there is still a need to focus on irrigated plantations that seek efficiency in the use of water (WUE). Therefore, it needs to be complemented.

2) Among the defense mechanisms, as well as antioxidants and phytohormones, were well explored. However, it lacked to include in a more detailed way an important mechanism that are the osmoregulators as well as proline.

3) Include a sub-item with soil and plant management that minimize water stress as well as application of agricultural gypsum (to neutralize subsurface aluminum and increase root volume), no-till to retain water, super-absorbent polymers, application of silicon, growth-regulating microorganisms , management that complement the genetic gain for this character.

4) Somewhere in the review, point out why drought and salinity tolerance mechanisms are common.

5) The improvement part involving TS and USA should begin, discussing whether the fact that these traits are quantitative, with many genes involved with little effect and high environmental influence, and why this makes the selection process difficult.

6) Among the breeding methods described, insert and discuss recurrent selection which are repeated processes of selection and recombination, thus concentrating favorable genes, which are many for quantitative traits, but with little effect.

7) Describe at least one existing phenotyping platform, preferably under field conditions, for TS and USA that can support breeding programs.

8) Include a more detailed sub-item on proteomics and TS/USA.

9) Include a more detailed sub-item on climate change and TS/USA.

10) Another convenient sub-item would be Phenomics (automated phenotyping) for TS and USA. This is because a phenotyping that is deficient in terms of efficiency, that is not high-troughput and that is destructive, would strongly limit the results of genomics, even if advanced.

11) Among the genes described in addition to DREB, include AREB genes.

12) When talking about waxiness, it would be good to link this character to a decrease in leaf temperature.

13) When discussing Genetic diversity and molecular markers, one must first explain the difference between dominant and codominant markers and the efficiency of molecular characterization.

14) Only 34% of bibliographic citations were published in the last 4 years. Please check for newer publications, considering that this is a review with an emphasis on the molecular area.

15) Line 25- Keywords like Molecular Breeding and Tolerance, already in the title.

16) Line 16 – Describe other concerns.

17) Line 65 – When you say In both cultivars, make it clear which ones they are.

Reviewer 2 Report

Authors did a very good job in putting together all current knowldge regarding diverse molecular approaches in chickpea breeding. Some references could be eliminated as knowledge is already stablished and there is no need to cite each and every sentence.

Authors must be careful because the title presented by journal Molecular approaches in the development of drought tolerance in chickpea is different from the one mentioned in the manuscript Molecular Breeding and Drought tolerance in chickpea.

The authors are invited to connect harmoniously the sentences. Example: Drought tolerance indicator in a crop should be directly connected with yield [214,215]. However, yield traits of crops have been found sensitive under drought stress [216].

What is the conclusion?

Many other examples are found.

Authors are invited to develop the conclusions and future work to be done by scientific community in this context. By this the manuscript doesn’t look merely descriptive and give some insight for current generation of scientists to prograde to a common good.

Reviewer 3 Report

1. I believe that the word "drought" should be found in keywords.

2. Genetic diversity can provide very pleasant surprises for plant breeding researchers. I myself have patented 4 varieties of vegetables from landraces existing at small farmers and in peasant gardens. Therefore, it would be very interesting to introduce some data on possible chickpea landraces with drought resistance. India is a country of very generous specific and genetic diversity. Molecular breeding, while simpler in some respects, also carries certain risks. We know about the failure of genetically modified crops in India with Bt cotton since 1998. Also, the European Union is very reserved about the cultivation of genetically modified plants. Therefore, landraces can be a less expensive and safer solution. If such data do not exist, I suggest the authors consider this aspect for future investigations.

3. Some data related to the diversity of patented varieties in India and possibly other large chickpea growing countries should be included in the paper. In the paper, only a few varieties are referred to, without having the overall picture of the genetic diversity.
